# Learning the Language of Social Media: A Comparison of Engagement Metrics and Social Media Strategies Used by Food and Nutrition-Related Social Media Accounts

**DOI:** 10.3390/nu12092839

**Published:** 2020-09-16

**Authors:** Amy M. Barklamb, Annika Molenaar, Linda Brennan, Stephanie Evans, Jamie Choong, Emma Herron, Mike Reid, Tracy A. McCaffrey

**Affiliations:** 1Department of Nutrition, Dietetics and Food, Monash University, Notting Hill, Melbourne 3168, Australia; amy.barklamb@monash.edu (A.M.B.); annika.molenaar@monash.edu (A.M.); steph.evans@sky.com (S.E.); waiyipchoong@gmail.com (J.C.); emherron555@gmail.com (E.H.); 2School of Media and Communications, RMIT University, Melbourne 3000, Australia; linda.brennan@rmit.edu.au; 3Department of Nutritional Sciences, University of Surrey, Guildford, Surrey GU2 7XH, UK; 4Nutrition Innovation Centre for Food and Health (NICHE), Ulster University, Coleraine, Northern Ireland BT52 1SA, UK; 5School of Economics, Finance and Marketing, RMIT University, Melbourne 3000, Australia; mike.reid@rmit.edu.au

**Keywords:** social media, Facebook, Instagram, nutrition, health promotion, young adults

## Abstract

Health promoters have been unable to reach and engage people on social media (SM) to the extent that food industry brands and lifestyle personalities have. The objective of this study was to identify the SM post strategies associated with higher engagement in nutrition and food-related posts using a retrospective content analysis. The six most engaging posts from both Facebook and Instagram’s 10 most successful nutrition and food-related accounts were analysed across four fields. Subjective and objective post strategies were coded on 736 posts, and associations with engagement were explored using the Least Absolute Shrinkage and Selection Operator (LASSO). Lifestyle personalities recorded the highest absolute engagement, while health promoters recorded the highest engagement relative to follower count. Strategies associated with higher Facebook engagement included using hashtags and prompting engagement through announcements, while on Instagram, higher engagement was associated with higher caption counts, providing health information links, prompting engagement through strategies that require an action, and using humorous strategies. Strategies associated with lower Instagram engagement included reposted content, general encouragement to eat strategies, encouragement to exercise strategies, not inducing any emotion/hedonic sensations, and providing a negative tone. Health promoters should adapt SM posts to the different SM platforms and utilise strategies associated with higher engagement to engage with their audience on SM.

## 1. Introduction

Health communication has been irrevocably changed with the arrival of social media platforms (e.g., Facebook, Instagram), which has facilitated the rise of a ‘platform society’ whereby societies are being formed and shaped on these platforms [1]. Social media refers to the reciprocal, web-based communication channels that, unlike traditional media, facilitate interactions and networking [2,3]. Globally, there are over 3.4 billion active users, allowing messages to reach the broader population and target individuals with common outcomes [2,4,5]. Social media success is often measured through reach, impressions, and engagement. Social media engagement can be defined as the total interactions on a social media post [6]. Interactions include social media users’ reactions (e.g., clicking on different emoji options on social media platforms, including like, love, haha, etc.), comments on the post, and sharing the post with their followers. Reach can either refer to the number of unique social media users that see an individual post (post reach) or all the posts on a social media page (page reach), whereas impressions refer to the total number of times a social media post is displayed on people’s screens, i.e., the post may be displayed on the same user’s screen multiple times [6].

Health promoters have begun to utilise social media for nutrition promotion as it allows for learning opportunities, support from like-minded users, community development, and the opportunity to set goals and review one’s progress [7,8]. Individuals who receive health messages via private Facebook discussion groups have been shown to have significantly greater reductions in body mass index, waist circumference, total cholesterol, and low-density lipoprotein cholesterol than those who do not [9]. Conversely, social media can foster an environment for poor dietary behaviours and body dissatisfaction [10]. Individuals can present heavily staged and edited content that portrays a ‘perfect lifestyle’, that may not reflect their true self [11]. Consequently, viewing or engaging with these ‘idyllic images’ can negatively affect body image and contribute to dieting/restricting of food or overeating and a greater pursuit of external validation. In addition, there is an abundance of food and nutrition content on social media with often conflicting messages, which may be due to the difficulty in interpreting the results of different studies with methodological differences [12,13]. When individuals are exposed to conflicting messages, it can lead to doubt surrounding the messenger’s credibility, ambiguity aversion, and undesirable dietary behaviours [12,13,14]. This is particularly concerning for the young adult population (18–24-year-olds). Young adults are noted to have an interest in specialist and novel foods, which are often considered healthy despite the actual nutritional value of the organic food [15,16]. They also have increased access to a plethora of recipes and information through social media, and whilst this may be expanding their eating and cooking habits [17], the differing nutritional value of these recipes may not assist in distilling conflicting nutrition information online [18]. 

Content on food, health, and nutrition on social media is dominated by Food Industry Brands (FIB) and Lifestyle Personalities (LP). Social media serves as a channel to increase commercial gains with little regulation over what is posted [19]. FIB are those who mass-market the food and beverage fast-moving consumer goods (FMCG), of which many are energy-dense and nutrient-poor [20]. Armed with large marketing budgets, they can reach the masses with messages often aimed at how products can enhance consumer attributes (e.g., heightened personal appeal) and normalising them within social contexts and youth culture [19]. Digital marketing has been shown to positively affect both attitudes towards and purchase intentions for unhealthy FMCG in young adults [21]. LP, commonly referred to as ‘social media influencers’, are independent third-party endorsers [22]. They rely on their online presence and success to earn an income, often at the cost of providing evidence-based advice [2,23]. Social media influencers have been found to promote alcohol consumption, despite having a young and potentially easily influenced audience, often adolescents and young adults [24]. LP frequently promote diet products such as the popular ‘FlatTummyCo appetite suppressant lollipops’ [25]. However, ‘satieral’, the active ingredient in these lollipops, has recently been found to have no effect on energy intake, hunger, or cravings [26]. Promotion of these non-evidence-based products likely feeds into the online cultural acceptance of extreme dieting, which may lead to body image concerns [27].

Health Organisations (HO) and Nutrition Professionals (NP) have begun to share their evidence-based messages online, sometimes in an attempt to counter FIB’s and LP’s potentially harmful messages. HO are Government and non-government third-party organisations, while NP are individuals with tertiary training in nutrition who traditionally have presented evidence-based messages in face-to-face settings [28,29,30]. Both (referred to as health promoters) present their messages with the intention of improving the health and wellbeing of society, rather than monetary gains [31]. However, they are often limited by resources, such as time and money, resulting in their outreach being significantly lower than FIB and LP [32,33]. Additionally, they suffer from the notion that they are no longer perceived as ‘experts’ online [34]. For the young adult population, evidence suggests that they commonly perceive authoritative bodies based on popularity rather than accuracy [34]. In order to reach and engage young adults with evidence-based messages, there is a need to identify what strategies engage this population. 

Previous research has focused on social media content beyond nutrition or has not included NP or extensive analysis of subjective data [3,27]. Research suggests that most users evaluate online message credibility through methods that require the least amount of psychological exertion and time (e.g., visual design) [35]. If engagement is based on the visual appeal of the social media posts rather than the emotion or tone of the post, recommendations to resource-restrained health promoters must consider this. The aim of this exploratory social media content analysis is to identify the strategies associated with higher engagement in food and nutrition-related posts. This research is an extension of Klassen et al. [36], which was a novel study using content analysis of social media posts in the context of nutrition. The rationale for extending the previous study was due to the ever-changing nature and importance of social media, particularly for nutrition, and the need to determine if there are differences in successful engagement strategies over time. The current study extends on the previous, with the novel inclusion of accounts run by NP and a detailed analysis of subjective post content strategies. This study compared strategies between Facebook and Instagram, objective and subjective data, and summer and winter posts. 

## 2. Materials and Methods

### 2.1. Design

This study was part of the broader Communicating Health study, which is a four-year multiphase study that aims to explore the utilisation of social marketing techniques to understand how to engage with young adults regarding health and nutrition, particularly on social media. The findings from the different phases will be used to inform the development of a toolkit to inform health promoters on how to utilise social marketing techniques and to better communicate with young adults via social media. The full protocol of the Communicating Health study has been published elsewhere [37]. This current study formed part of the formative research phase of the Communicating Health study and involved a retrospective content analysis to identify which social media strategies engage users in food and nutrition-related posts. Posts were selected from accounts within four food and nutrition-related fields: FIB, HO, LP, and NP, herein referred to as fields. 

### 2.2. Procedure

Data were collected during summer (19 February to 5 April 2018, southern hemisphere) and winter (25 June to 29 July 2018). Socialbakers© (Prague, Czech Republic), an online social media analytic company, identified the 10 most popular Facebook profiles amongst Australian users for each field. The researchers chose to focus on Australian-based social media accounts or accounts that had a large Australian following due to the inclusion of NP. We needed to verify the NP qualifications based on Australian tertiary education providers, the Nutrition Society of Australia register [38], and the Dietitians Association of Australia as there are no international standards [39].

Popularity was determined based on the number of Australian followers each profile had at the time of data collection. The Klassen et al. [36] filters for searching through Socialbakers© were applied: ‘FMCG’ for FIB, ‘society’ for HO, and ‘celebrities’ or ‘lifestyle’ for the LP field. International FIB and LP accounts were included if they had an Australian social media presence whereas HO were Australian-based. As NP were an addition to this study, filters were not available in Socialbakers©. Instead, NP were identified by internet searches and had to have a bachelor’s degree or a higher degree in a nutrition field, with three or more years’ experience in the industry. This was based on the information provided on their social media profile or website at the time of data collection. 

Once Facebook profiles were identified, Socialbakers© was then used to identify each profile’s six most popular posts during the data collection period. Popularity was determined using the highest engagement with users: the total interaction count (sum of comments, reactions, and shares). Comments refer to users leaving responses on posts, reactions refer to users clicking any of the ‘like’, ‘love’, ‘haha’, ‘wow’, ‘sad’, and ‘angry’ emojis Facebook provides, while shares refer to users publicly sharing the post amongst their own followers. To determine the top Instagram posts, Socialbakers© was used to see if the same Facebook profiles had an equivalent Instagram page. If there was either no Instagram profile or Socialbakers© did not provide profile data, they were excluded from the Instagram analysis only (*n* = 23 profiles). The six most popular posts per profile were identified; however, the total interaction count was the sum of ‘loves’ (Instagram’s equivalent to Facebook’s reactions, herein referred to as reactions) and comments. Due to the inconsistent use of reactions on Facebook, which has six ways to react, and Instagram, which has one possible reaction type, only total reactions are presented. All data generated by Socialbakers© were downloaded, screenshots of the posts were taken, and ID numbers were allocated to each post. A total of 960 potential posts were identified to be collected (Figure 1).

Strategies were identified using an adapted coding framework which was iteratively refined during the coding process. Appendix B provides this coding framework, with overall strategies and their subcategories being defined. A consensus method was used whereby each post was reviewed by two authors (E.H and J.C—summer posts, A.M.B and S.E—winter posts). Where there were any discrepancies, the authors discussed, and if a consensus was not reached, A.M. and T.A.M provided input. The authors who coded the posts were chosen as they were themselves young adults. 

### 2.3. Statistical Analysis

Statistical analysis was performed using IBM SPSS Statistics 25 [IBM Corporation Armonk, NY, USA] and R [R Foundation for Statistical Computing, Vienna, Austria]. Kruskal–Wallis ranks test was used to determine differences between the four fields, with Dunn’s test indicating where any differences occurred. A Chi-square independence test was used to determine differences between strategies within the coding framework. Significance was set at *p* < 0.05 unless Bonferroni correction was necessary for multiple comparisons. 

To determine which strategies influence engagement the most, LASSO (Least Absolute Shrinkage and Selection Operator) regression was undertaken [40]. LASSO regularisation helps to determine which variables should be included in the regression as incorrect selection can lead to overfitting and misconstrued results (overfitting—the tendency for the model to fit extremely well to the current data but not future observations) [41].
LASSO = ^min^(sum of the squared residuals) + (λ × sum of |magnitude of coefficients|)(1)

LASSO differs from traditional regression by introducing a ‘penalty term’ to the least-squares method (Equation (1)), to reduce the overall variance of the model. This penalty is calculated by ‘lambda times the sum of the absolute values of the coefficients’. It determines which predictor variables are important by shrinking the coefficients of those that are not to zero. 

The Friedman et al. [42] ‘glmnet’ package was implemented in R to perform the LASSO. A matrix of data was created that included all recoded data found in the coding framework (Appendix B; with the exception of video duration). This matrix was split into two equal parts: testing and training data. Using the training data, cross-validation was performed to determine the optimal value of lambda. Here, data were divided five-fold, four of which were fitted to the LASSO equation over a range of lambda values. The results were compared against the remaining fold to establish a prediction error. The value of lambda that resulted in the smallest prediction error was used as the ‘optimal lambda’ and placed back into the LASSO equation but, this time, using the testing data. The coefficients of each variable were determined, with those that were not significant being sent to zero. This process was performed on both Facebook and Instagram data separately. For more details on how to perform LASSO in R, please refer to Hastie et al. [43]. 

### 2.4. Ethics

Ethics was approved by the Monash University Human Research Ethics Committee (no.13792). Social media account names were removed and replaced with ID numbers to ensure confidentiality. None of the authors were affiliated with nor in contact with any of the accounts analysed. 

## 3. Results

### 3.1. Social Media Engagement by Platform and Field

Of the identified posts (960), 736 posts were collected and analysed. Posts were missing due to either less than six being uploaded, there was no equivalent Instagram profile, or posts were deleted between the Socialbakers© collection and the coding framework analysis. Posts with incomplete data were due to either Socialbakers© not producing the relevant objective data or posts being deleted before analysis was complete. Whilst it appeared that most posts were distributed using organic methods rather than paid advertising, these data were only available for 581 posts (79%, Table 1) with missing data more likely to be from NP and HO; thus, further analysis was inappropriate.

#### 3.1.1. Objective Strategies

By virtue of the identification process, all fields primarily used Facebook, with 58.4% of posts coming from the platform (Table 1). LP reported significantly more Facebook followers than HO (*p* = 0.001) and NP (*p* < 0.001); however, when adjusting for followers, HO and NP reported significantly higher total relative interactions than FIB (*p* < 0.001) and LP (*p* < 0.001). This was similarly seen on Instagram, where although LP recorded the highest median followers, HO and NP had significantly higher relative total Instagram interaction (*p* < 0.001). Only 55.3% of the summer profiles (both platforms) were also identified in the winter collection; thus, seasonal comparisons were deemed inappropriate. Regardless of the platform, the majority of posts were photos (Facebook, *p* < 0.001. Instagram, *p* = 0.01), although videos were more frequent on Facebook than Instagram (predominantly from FIB and LP). Text-only posts were mainly seen by HO and NP (*p* < 0.001). HO had significantly higher caption counts than FIB (*p* < 0.001), LP (Facebook; *p* = 0.01. Instagram; *p* < 0.001), and NP (Instagram only; *p* < 0.001). However, the majority of posts (86%) included a caption count of fewer than 100 words. Links to other social media channels were the most frequently reported, whereas health information links were rare and primarily came from HO and NP on Facebook (*p* < 0.001) and HO only on Instagram (*p* = 0.01). Unsurprisingly, the promotion of a product was mainly seen by FIB and LP and was performed more often on Facebook than Instagram (Facebook; *p* < 0.001. Instagram; *p* < 0.001). 

Food was shown on 48.8% of Facebook posts, with FIB having the highest proportion (89.6%), followed by NP (51.8%) then LP (32.0%) and HO (25.0%, Appendix A). Of these, FIB primarily displayed non-core foods (96.5%) while core foods made up the majority of NP (78.9%) and HO (75.0%) food posts. Interestingly, LP had relatively similar proportions of core and non-core foods. Instagram had a slightly higher proportion of posts containing food (59.5%), with core foods more frequently shown across all four fields. As expected, HO and NP primarily used health-promoting content on both platforms while FIB rarely used it (*p* < 0.001, Appendix C). LP reported similar proportions of health-promoting content to other content (e.g., jokes, memes, etc.). In core food posts, almost all had health-promoting content, regardless of the platform, while on the non-core food posts, health was rarely promoted. 

Sixty-three percent of Facebook posts (*p* = 0.002, Appendix C) and 59% of Instagram posts (*p* = 0.004) included promoting-engagement strategies, mainly announcements, and require a response. Relationship building was less common (Facebook, 49%. Instagram, 44%), with external content and replying to the audience mostly used (*p* < 0.001). Although real-world tie-ins were not overly common (Facebook, 37%, *p* < 0.001. Instagram, 43%, *p* = 0.01), HO were the biggest users. Encouragement to eat was seen on most posts (Facebook, 53%. Instagram, 60%, *p* < 0.001), with general encouragement and food shown to be the most popular strategies. Within the general encouragement strategy, balanced food choices were mainly seen by NP and very rarely by LP or FIB (Appendix A). A large proportion of the non-core food posts (95.5%) reported an encouragement to eat strategy. All fields rarely used any encouragement to exercise (Facebook, 18%. Instagram, 12% (*p* < 0.001)) strategies.

#### 3.1.2. Subjective Strategies

Trying to induce emotion or hedonic sensations was ubiquitous on both Facebook (90%) and Instagram (94%, Appendix D, *p* < 0.001), with most fields attempting this in a positive way, particularly for core foods. When core foods were coded as negatively inducing emotion or hedonic sensations, this was only by HO and NP. Interestingly, most Facebook non-core food posts were also classified as positive, even those from HO (50%) and NP (83%), while all Instagram non-core food posts came under a positive classification. The majority of posts, across all platforms and fields, had a positive tone or less often a neutral tone. Association with success strategies were also uncommon (Facebook; 14.0%. Instagram; 13% *p* < 0.001). On Facebook, FIB mainly used strategies that related to a product (e.g., price promotion, product launch, etc.) followed by visually-appealing strategies (*p* < 0.001). HO, LP, and NP were more likely to provide relatable content (e.g., stories, friendship, etc.). Furthermore, HO and NP commonly used statistics/facts, particularly on Facebook posts. Instagram reported similar results (*p* < 0.001); however, visually-appealing strategies were the most frequent for FIB and HO. 

### 3.2. Multivariable Linear Regression: Objective Strategies

Two LASSO models, one for each platform, were developed to explore the relationship between engagement, platform, and the types of strategies used (Table 2). On Facebook, strategies that were associated with higher engagement included the use of hashtags and announcements compared to not prompting engagement strategies. All other strategies in this model were reduced to zero, and thus not significant. 

On Instagram, strategies that were associated with higher engagement included longer caption counts, providing links to health information, presenting other (e.g., jokes, images of friends) content over health-promotion content, using strategies that required an action, and providing humorous strategies as opposed to providing personal/relatable content strategies. Strategies associated with lower Instagram engagement included reposting content compared to uploaded content, encouragement to eat and exercise, not inducing any emotion/hedonic sensations, and using a negative tone. 

## 4. Discussion

This exploratory content analysis found both objective and subjective social media strategies with higher engagement differing by platform and field. LP had the highest overall engagement (followers, total Facebook, and Instagram interactions), while HO had the lowest. However, adjusting for the number of followers resulted in HO and NP receiving more relative engagement than LP and FIB. This indicates that while health promoters are engaging with their followers, they do not, however, have the page reach that FIB and LP do. Relative interactions were higher on Instagram than Facebook, consistent with new research that highlights a shift away from Facebook towards Instagram as the preferred platform by young adults [44]. Future interventions must identify and consider this shift, disseminating messages on platforms that resonate with the target group first using strategies to acquire an audience and then strategies that engage and retain their acquired followers. 

Based on the current findings, recommendations for health promoters’ Facebook and Instagram use are provided in Table 3. 

Similar to the previous analyses by Klassen et al., LP continue to have the highest absolute engagement on both platforms while HO have the least. Although Hitlin et al. [45] used different fields, their report highlighted that Facebook primary pages (similar to the current LP) received more engagement than multiplatform organisations (similar to the current FIB and HO). Photos are consistently the most popular format used [45,46], in some cases with more success than videos, although this was not seen in the present study [47]. HO have continued to use more prompts for engagement and real-world-tie-in strategies, links to health information, and a more neutral tone than other fields [46]. Previous research found sponsored health-related content with a link or a photo were more successful on Facebook after adjusting for reach than videos [48]. In addition, emotional communication strategies received lower engagement than instructive/call-to-action communication strategies [48]. 

Showing people exercising was not associated with higher engagement on either platform, which is dissimilar to previous research on social media ‘Fitspiration’ content, where this type of post was seen as useful and motivational [49]. Fitspiration content was commonly accessed by young adults through LP, such as personal trainers and athletes, as well as ‘everyday people’, including friends and peers [49]. Furthermore, participants perceived qualified fitness experts and individuals who were relatable as trustworthy sources of health information [49]. However, fitspiration also brought about pressure to meet a ‘healthy ideal’ [49]. 

Klassen et al. identified that Facebook posts that featured people, relatable content, pop culture, stories, and weight loss content were associated with lower engagement [36]. In the current results, only weight loss content (under the grouping of health-promoting content) was associated with lower engagement. Pop culture (under the grouping of humorous strategies) was, in fact, associated with higher engagement in the present study [45,46]. Posts that provided links to purchasable items were seen in the previous analysis to increase engagement on both platforms, whereas in the present study, they were not identified to have significant impact on engagement. Consumers, particularly young adults, appear to be more conscious about the health and sustainability of their food choices than previous generations, and are often well-informed and purposeful about the food products they buy [50]. Young adults may be cognisant of which posts are sponsored, and when promoting food products, FIB may be more credible than LP [51]. Online user-generated content (e.g., reviews) relating to fast-food brands have been shown to positively affect brand commitment and brand loyalty but may not have a significant effect on brand trust or satisfaction [52]. However, our findings indicated that audience-generated content (under the grouping of external content) coming from FIB was not associated with increased engagement, although we did not specifically assess the reviews or audience-generated content relating to the external content of a brand. 

Klassen et al. also identified that weight loss content and hashtags on Instagram posts were associated with reduced engagement, yet the present results only found the former to be true. Previous research has indicated that weight loss online communities can be a source of support, relatability, and connectedness during weight loss journeys and the interaction with weight loss content amongst these communities is particularly high for success stories [53]. Relatable content was not significant in the current results; however, it has previously been associated with higher Instagram engagement. These differences between the current analyses and the analyses by Klassen et al. may be explained by the fact we split our coding framework and analyses into objective and subjective strategies, as well as the inherent time differences and differing social media profiles analysed. Furthermore, the findings from the subjective strategies are relevant for a young adult population given the position of the authors (A.M.B., S.E., E.H, and J.C), whereas only one of the authors of Klassen et al. was a young adult. 

Through giving individuals the opportunity to interact with like-minded individuals pursuing similar goals, social media creates a sense of belonging and provides social support and attachment to others [7,8]. When using social-media-based interventions, these benefits have shown to improve nutritional status, such as improved weight management and dietary fat consumption [54,55]. These benefits are not isolated to only nutrition education, with improvements being seen in smoking cessation rates, quality of life scores, reduced medical visits, along with reduced rates of suicidal ideation and improved symptoms of cancer-related depression [56,57,58,59]. However, these benefits appear to be isolated to private discussion groups rather than public social media posts. Young adults have shown interest in partaking in online healthy eating discussions but are restricted by the social stigma surrounding online weight-related talk [8]. There is a need for future interventions to disseminate messages into existing social networks to optimise social support and engagement for young adults.

The current research is strengthened by the novel inclusion of both subjective data and an NP field. However, we were unable to determine demographics (i.e., age or gender) of the followers or overall reach or impressions of the posts, as these were not available at the time of data collection and could only be provided by account administrators. This limited the ability to identify where the interactions were coming from and if the posts resonated specifically with young adults. Young adults were selected to code the data as they are able to have a reasonable understanding of what engages their peers. Furthermore, the research team has a background in nutrition and may code social media posts differently to those with no nutritional background. The results are also limited by the fast-paced nature of social media, where in the future, a new social media platform may be developed, and the next young adult generation may interact with it in different ways. Therefore, it is essential that future research continues to update and amend recommendations through similar studies, particularly from the position of the target group. Additional consideration could be given to whether the profiles are based on an organisation or individual and are for commercial or non-commercial gains as this may impact engagement. As data analysis only occurred on the caption, any disclosure of sponsorship in the comments was not identified. By law, Australian accounts must disclose all advertising and marketing communication [60]. However, as the code does not stipulate where the ‘#ad’ requirement is placed, the research team often saw it appear in posts’ comments. Therefore, the current results may not represent the true proportion of sponsorship strategies. 

In parallel to these findings, our Communicating Health project has provided an understanding of the need for nuanced messaging that avoids using the rhetoric of the ‘good and ‘bad’ approach to food and health [61] and responds to how young adults describe health [62]. Thus, future studies should track whether the implementation of the provided recommendations resulted in increased engagement metrics on their platform. It is acknowledged that whilst strategies may be associated with higher engagement, this does not mean that users will engage in the behaviours being promoted. The continued application of social marketing techniques in food and health will provide a useful roadmap of successful communication strategies for social media. Potential future work could explore the relationship between user’s perceptions of influencers’ authenticity and how this affects whether they partake in the messages presented. 

## 5. Conclusions

Social media posts in this exploratory content analysis had higher engagement with differing objective and subjective social media strategies depending on the platform, field, and season. Strategies such as external health information, prompting engagement with followers, and using a positive tone were associated with higher engagement on Instagram, whereas on Facebook, using hashtags and providing announcements were associated with higher engagement. This study enhances the work previously performed by classifying strategies into objective and subjective approaches. Our findings reiterate the notion that creators should adapt posts to the platform being used. While health promoters do not have the same reach (number of people seeing posts) as FIB and LP, their audiences are highly engaged with their social media posts. Thus, examining their audience profile in line with their organisation objectives and tailoring social media posts using our recommendations may assist with increasing reach. 

## Figures and Tables

**Figure 1 nutrients-12-02839-f001:**
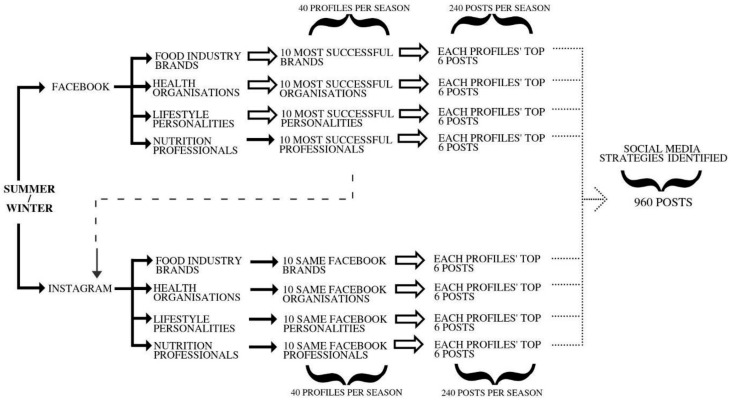
Graphical representation of study design. Solid arrows indicate steps where decisions were made by the researchers, i.e., including Facebook and Instagram rather than other platforms; including Food Industry Brands, Health Organisations, Lifestyle Personalities, and Nutrition Professionals as groups of interest. The top 10 NP profiles were selected by researchers based on criteria specified in the methods. The dashed arrow indicates where the Facebook data collection informed the Instagram profiles. Hollow arrows indicate that the data to the right was collected via Socialbakers©. Dotted arrows indicate the collection via the coding framework.

**Table 1 nutrients-12-02839-t001:** Engagement and format metrics by platform and field.

	Facebook Posts					Instagram Posts				
	Overall *n* = 430	FIB *n* = 96	HO *n* = 93	LP *n* = 131	NP *n* = 110	Overall *n* = 306	FIB *n* = 75	HO *n* = 51	LP *n* = 110	NP *n* = 70
**Engagement**	*					*				
Fans ^1^	275,365 (22,651; 3,910,357)	1,054,041 ^a^ (645,649; 16,682,426)	22,070 ^b^ (13,202;30,714)	3,910,357 ^a^ (1,228,447; 20,335,695)	24,310 ^b^ (15,552; 44,854)	39,478 (8,189; 581,713)	42,452 ^a^ (23,613; 270,839)	2,841 ^b^ (1,826; 5,852)	2,388,095 ^c^ (269,317; 6,450,564)	13,142 ^d^ (5,257; 28,402)
**Total interactions (Relative) ^1^**	0.13 (0.05; 0.33)	0.02 ^a^ (0.003; 0.08)	0.21 ^b^ (0.09; 0.62)	0.14 ^c^ (0.08; 0.27)	0.29 ^b^ (0.1; 1.03)	1.73 (1.09; 3.11)	1.55 ^a,b^ (0.94; 3.11)	2.73 ^c^ (1.61; 3.67)	1.42 ^a^ (1.0; 1.97)	2.47 ^b,c^ (1.4; 3.58)
Comments (Relative) ^1^	0.02 (0.004; 0.06)	0.1 ^a^ (0.002; 0.07)	0.02 ^a^ (0.004; 0.09)	0.01 ^a^ (0.01; 0.04)	0.03 ^a^ (0.01; 0.1)	0.04 (0.01; 0.1)	0.03 ^a^ (0.01; 0.13)	0.05 ^a,b^ (0.01; 0.19)	0.02 ^c^ (0.01; 0.04)	0.08 ^b^ (0.03; 0.18)
Reactions (Relative) ^1^	0.10 (0.05; 0.22)	0.03 ^a^ (0.01; 0.07)	0.16 ^b^ (0.08; 0.5)	0.1 ^c^ (0.06; 0.16)	0.21 ^b,c^ (0.09; 1.22)	1.70 (1.06; 2.95)	1.52 ^a,b^ (0.94; 2.98)	2.59 ^c,d^ (1.6; 3.46)	1.39 ^a^ (0.99; 1.93)	2.28 ^b,d^ (1.3; 3.44)
Shares (Relative) ^1^	1.01 (0.11; 6.52)	0.13 ^a^ (0.02; 0.42)	3.51 ^b^ (1.22; 12.26)	0.78 ^c^ (0.11; 6.27)	2.14 ^b,c^ (0.41; 8.5)	-	-	-	-	-
**Season ^2^**	**									
Summer	214 (49.8)	59 (61.5)	33 (35.5)	59 (45.0)	63 (57.3)	165 (53.9)	35 (46.7)	24 (47.1)	60 (54.5)	46 (65.7)
Winter	216 (50.2)	37 (38.5)	60 (64.5)	72 (55.0)	47 (42.7)	141 (46.1)	40 (53.3)	27 (52.9)	50 (45.5)	24 (34.3)
**Post Reach ^2^**										
Organic	268 (60.0)	43 (44.8)	69 (74.2)	96 (73.3)	50 (45.5)	211 (69.0)	70 (93.3)	25 (49.0)	95 (86.4)	21 (30.0)
Paid advertising	96 (22.3)	52 (54.2)	12 (12.9)	28 (21.4)	4 (3.6)	16 (5.2)	5 (6.7)	2 (3.9)	9 (8.2)	0 (0.0)
Unknown	76 (17.7)	1 (1.0)	12 (12.9)	7 (5.3)	56 (50.9)	76 (25.8)	0 (0.0)	24 (47.1)	6 (5.5)	49 (90.0)
**Format ^2^**	***					**				
Photos	224 (52.1)	53 (55.2)	52 (55.9)	65 (49.6)	54 (49.1)	249 (81.4)	63 (84.0)	44 (86.3)	78 (70.9)	64 (91.4)
Videos	128 (29.8)	42 (43.8)	17 (18.3)	56 (42.7)	13 (11.8)	57 (18.6)	12 (16.0)	7 (13.7)	32 (29.1)	6 (8.6)
*Duration* ^1^	0:37 (0:09; 2:00)	0:06 (0:06; 0:11)	0:30 (0:20; 0:53)	1:47 (0:45; 2:54)	1:05 (0:56; 2:10)	0:44 (0:10; 1:00)	0:10 (0:07; 0:12)	0:02 (0:02; 0:02)	00:58 (0:44; 1:00)	-
Text only	78 (18.1)	1 (1.0)	24 (25.8)	10 (7.6)	43 (39.1)	0 (0)	0 (0)	0 (0)	0 (0)	0 (0)
**Post origin ^2^**	**									
Uploaded	386 (89.8)	96 (100)	79 (84.9)	117 (89.3)	94 (85.5)	240 (78.4)	54 (72.0)	40 (78.4)	87 (79.1)	59 (84.3)
Reposted	44 (10.2)	0 (0)	14 (15.1)	14 (10.7)	16 (14.5)	66 (21.6)	21 (28.0)	11 (21.6)	23 (20.9)	11 (15.7)
**Shows people ^2^**	189 *** (44.2)	15 (16.0)	51 (54.8)	82 (62.6)	41 (37.3)	125 *** (40.8)	16 (21.3)	24 (47.1)	57 (51.8)	28 (40.0)
**Caption count ^1^**	32 * (17; 65)	21 ^a^ (14; 29)	53 ^b^ (32; 83)	27 ^c^ (16; 78)	44 ^b,c^ (16; 88)	29 * (15; 62)	17 ^a^ (11,24)	56 ^b^ (41; 72)	24^c^ (13; 58)	52 ^b^ (26; 100)
**Hashtags used ^2^**	98 *** (22.8)	12 (12.5)	48 (51.6)	23 (17.6)	15 (13.6)	185 *** (60.5)	48 (64.0)	43 (84.3)	48 (43.6)	46 (65.7)
**Links to other SMC ^2^**	274 ** (63.7)	21 (21.9)	76 (81.7)	96 (73.3)	81 (73.6)	133 (43.5)	29 (38.7)	24 (47.1)	45 (40.9)	35 (50.0)
**Promotion of a product ^2^**	181 *** (42.1)	74 (77.1)	27 (29.0)	63 (48.1)	17 (15.5)	82 *** (26.8)	48 (64.0)	8 (15.7)	20 (18.2)	6 (8.6)
**Links to health information ^2^**	47 *** (10.9)	0 (0)	21 (22.6)	2 (1.5)	24 (21.8)	3 ** (1.0)	0 (0)	3 (5.9)	0 (0)	0 (0)

* *p* < 0.05. ^a–d^ Values in rows and subtables not sharing the same subscript are significantly different using the Kruskal–Wallis test and post hoc Dunn’s test with Bonferroni adjustment for multiple comparisons. Tests assume equal variances; ** *p* < 0.01, Chi-square; *** *p* < 0.001; Overall categories are in bold (e.g., total interactions), with subcategories below (e.g., comments or shares). ^1^ Median (25th; 75th); ^2^ n (%); Abbreviations: ‘FIB’ food industry brands. ‘HO’ health-promoting organisations. ‘LP’ lifestyle personalities. ‘NP’ nutrition professionals. ‘Relative’, relative to followers (presented as %). ‘SMC’, social media channels. Overall strategies are in bold (e.g., post reach), with subcategories stated below (e.g., organic, paid advertising, unknown).

**Table 2 nutrients-12-02839-t002:** Least Absolute Shrinkage and Selection Operator (LASSO) regression coefficients.

	Facebook	Instagram
	Coefficients	Coefficients
Engagement (intercept)	0.279	2.850
Predictors		
**Fields**		
Food Industry Brands	−0.033	
Health Organisations	0.422	0.280
Lifestyle Personalities	Ref.	Ref.
Nutrition professionals	0.274	
**Origin of post**		
Uploaded content	Ref.	Ref.
Reposted content		−0.449
**Caption count**		0.584
**Hashtags used**		
No	Ref.	Ref.
Yes	0.0002	
**Links to health information**		
No	Ref.	Ref.
Yes		2.204
**Content type**		
Health promoting		Ref.
Other	Ref.	0.020
**Prompting engagement**		
None	Ref.	Ref.
Announcement	0.064	
Requires a response		
Requires an action		0.053
**Encouragement to eat**		
None	Ref.	Ref.
Encouragement to eat, general		−0.751
Encouragement to eat, specific		
Food shown		
**Encouragement to exercise**		
No	Ref.	Ref.
Yes		−0.645
**Emotion/hedonic sensations**		
Positively induces	Ref.	Ref.
Negatively induces		
None		−0.021
**Tone**		
Positive	Ref.	Ref.
Neutral		
Negative		−0.699
**Social media strategy used**		
Visually appealing		
Statistics/facts		
Relates to a product		
Personal content/relatable content	Ref.	Ref.
Humorous		1.815

Abbreviations: ‘Ref.’ reference category for overall strategies. Overall strategies are in bold (e.g., tone), with subcategories stated below (e.g., positive, negative). If the LASSO sent all variables within a strategy group to zero, the group was not presented in the table.

**Table 3 nutrients-12-02839-t003:** Recommendations for health promoters based on strategies used and strategies associated with higher engagement.

Facebook	Instagram
Start using hashtags on posts Continue prompting engagement with users, specifically through ‘announcements’ when appropriate	Try to upload content you have created yourself The larger the caption count, the better Provide more health links on posts Continue providing ‘other’ content amongst posts that aim to improve health, e.g., jokes or lifehacks Use strategies that ‘require an action’, e.g., tagging, sign ups, to prompt users’ engagement Consider engaging ways to encourage consumption of core foods or exercising, i.e., positive emotions, tone, and hedonic sensations Consider using ‘humorous strategies’ such as memes or pop culture

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
