# Peer review of "Learning the Language of Social Media: A Comparison of Engagement Metrics and Social Media Strategies Used by Food and Nutrition-Related Social Media Accounts"

_nutrients, 2020, doi:10.3390/nu12092839_

Round 1
Reviewer 1 Report
Thank you for allowing me to review this manuscript.
Overall, it is an interesting study that, by means of content analysis, identify the strategies associated with higher engagement on food and nutrition related posts on Facebook and Instagram.
However, I do have a few comments.
Lines: 95-96. Sentence is difficult to understand. “If post visual design receives higher engagement than emotion or tone, recommendations to resource-restrained health promoters must consider this.” Should it be:
If engagement is based on the visual appeal of posts rather than their emotion or tone…”
Line: 98. The authors refer to Klassen et al. several times in the manuscript and state that their own research is an extension of Klassen’s work. It would be good to include a motivation to why Klassen’s work is seminal/of importance (i.e. what is the rationale of extending this particular research?)
Lines: 137-138. Figure 1. The hollow and dotted arrows are explained. However, no explanation of the solid arrows is provided.
Lines: 141-142. The coding was conducted by several coders. Did they estimate inter-rater reliability? If so, how was this process carried out?
Lines: 159-160. Rephrase sentence from “All authors were neither affiliated…” to: “None of the authors were affiliated…”
Lines: 238-256. This passage is difficult to read as all the values and coefficients disrupts the flow of the text. I do not see a need to repeat the values already presented in the table. I encourage the authors to reduce the number of values/coefficients and focus on the significant results in the text.
Lines: 267-280. Same as above. Readability is impaired due to the repeated values/coefficients already presented in the table. I encourage the authors to focus on the significant aspects in the text.
I encourage the authors to revise and resubmit. Good luck!
Reviewer 2 Report
It is with great interest and enthusiasm that I read this manuscript on nutrition-related social media strategies. The manuscripts pertains to an emerging field of research that is highly relevant and necessary for the health practitioners that also engage in social media in an attempt to bring balance to a medium that shows a strong bias towards unhealthy dietary choices being displayed. In all, I really liked to angle and design of the study and I truly expect that many practitioners will resort to its results. The analysis framework is solid, and the dataset with 736 posts being analyzed is large enough. Below, I do list the comments or concerns I still have (a couple of them mere suggestions), but I do wish the authors good luck with the continuation of this line of research.
-Some of the more recent work on this topic, but situated more in the domains of communications science and psychology, could be a good add on to the introduction and general framework. For instance, in a recent special issue in Frontiers in Psychology (https://www.frontiersin.org/research-topics/9295/the-role-of-social-media-influencers-in-the-lives-of-children-and-adolescents#articles) there are some relevant references.
-The "four domains" for the analysis are not as clear to the reader, so it would be good to maybe frame this differently
-The analysis now looks at an aggregate type of social media post interactions, but I would also be interested in the trimmed down version of this, certainly for the largest subcategories of these interactions (e.g. likes).
-I also wonder whether the researchers are still able to see in there data what would be the total exposure of all posts. If, for instance, health organization messages are more often shared, then their total exposure might be bigger than what we now believe. However, it could also be that the total exposure is even more biased towards the more unhealthy commercial messages.
-Another issue that would be good to shortly address is whether the messages analyzed were organic messages only or whether they were also used as paid advertising messages, which would bias the results (because then the reach of these messages is partly bought rather than organic). Similarly, platform dynamics might also limit the organic reach of the four different groups of senders in different ways. It is quite plausible that the organic reach for the health organizations is much higher than the organic reach of the commercial pages and this can impact the data. So, please shortly address this.
-With regard to the food types in the messages, I wonder whether they can also be analyzed as core/non-core/mixed, to give a deeper understanding of the data.
-It is with regard to the analysis that I have the strongest reservations at this point and these boil down to two different things. First, I am really concerned about the stepwise regression procedure. As has been claimed often before (see e.g. Frank Harrell in 2001, but also countless articles and stats blogs), this is not a stable procedure for many reasons. Looking at the results table, I also have the feeling that this is true for the analyses here where some of the regression coefficients seem to load negatively where you expect them to be positive etc. I would be in favor of just presenting raw, uncorrected regression weights for each parameter on the one hand, and the results for a full regression model (maybe dropping some variables that seem peripheral anyhow) on the other hand.
Second, I expected also a focus on the different dynamics for each of the four types of senders. These four senders could be structured in two variables (organization versus person; commercial versus non-commercial) or kept as four different categories. But, crucially, the analysis could then look at interaction effects (like a moderation analysis) showing that certain characteristics have a different effect on the outcome depending on which sender type applies the message characteristics. Such an analysis seems to provide a deeper understanding to highlight where non-commercial parties should in fact refrain from the commercial tactics.
Reviewer 3 Report
This manuscript deals with an interesting topic and might represent a sound contribution. However, I would like to see some improvements/clarifications, in particular:
- Please state more clearly the objective of your paper in the Abstract.
- I would like to see more recent literature on young consumers/food industry/ social media. Some key studies were neglected, here are some of them:
Pham, T. H., Nguyen, T. N., Phan, T. T. H., & Nguyen, N. T. (2019). Evaluating the purchase behaviour of organic food by young consumers in an emerging market economy. Journal of Strategic Marketing, 27(6), 540-556.
Šerić, N. & Garbin Praničević, D. (2018) Consumer-Generated Reviews on Social Media and Brand Relationship Outcomes in the Fast-Food Chain Industry, Journal of Hospitality Marketing & Management, 27:2, 218-238
Savelli, E., Murmura, F., Liberatore, L., Casolani, N., & Bravi, L. (2019). Consumer attitude and behaviour towards food quality among the young ones: Empirical evidences from a survey. Total Quality Management & Business Excellence, 30(1-2), 169-183.
- Please revise the number of self-cited studies.
- Please provide more information on design of the study. On page 3/26, lines 104-107 you state: “This study was part of the broader Communicating Health Study which is a four-year study which aims to explore the utilisation of social marketing techniques to understand how to engage with young adults regarding health and nutrition, particularly on social media. The full protocol of the Communicating Health study has been published elsewhere”. Does this mean that the results of this manuscript have already been published elsewhere?
- Please explain why you focused on Australian users and provide more information about this research context.
- There are too many tables, please revise
- Please summarize your key findings in Conclusion.
Round 2
Reviewer 3 Report
Thank you for revising the paper, I believe you did a good work in addressing all my concerns.